# A Preliminary Study of Intravaginal Lactic Acid Gel (Canesbalance^®^) for Post-Episiotomy Healing: A Randomized Clinical Trial

**DOI:** 10.3390/healthcare13131581

**Published:** 2025-07-01

**Authors:** Dragos Brezeanu, Ana-Maria Brezeanu, Sergiu Ioachim Chirila, Vlad Tica

**Affiliations:** 1Faculty of Medicine, Ovidius University of Constanta, 900573 Constanta, Romania; brezeanudragos@gmail.com (D.B.); sergiu.chirila@univ-ovidius.ro (S.I.C.); vtica@eeirh.org (V.T.); 2County Clinical Emergency Hospital “Sf. Ap. Andrei”, 900591 Constanta, Romania

**Keywords:** puerperium, perineal pain, perineal wound, vulvodinia

## Abstract

**Background:** Episiotomy-related morbidity remains a substantial challenge in postpartum recovery, often affecting maternal quality of life. Given the crucial role of local microbiota and wound environment in perineal healing, intravaginal lactic acid gel emerges as a promising adjunctive therapy. **Objective:** To evaluate the effectiveness and safety of intravaginal lactic acid gel (Canesbalance^®^, Bayer) in enhancing scar healing and reducing pain following mediolateral episiotomy. **Methods:** In this single-center randomized controlled trial, 100 postpartum women with mediolateral episiotomy were allocated to either a treatment group receiving intravaginal lactic acid gel (Canesbalance^®^, Bayer) or a standard care group. Scar healing was assessed at 7 and 40 days postpartum using POSAS, VAS, and NRS scores; hematological parameters were also monitored. **Results:** The lactic acid gel group demonstrated significantly greater reductions in scar severity and pain scores over time (*p* < 0.05), with no adverse hematological effects. Effect sizes (Cohen’s d) were moderate to high for scar healing (d = 0.76) and pain reduction (VAS: d = 0.83; NRS: d = 0.79), indicating clinical relevance beyond statistical significance. **Conclusions:** Intravaginal application of lactic acid gel may offer a safe and effective strategy to enhance perineal wound healing and pain relief after episiotomy. Further large-scale studies are warranted to validate these promising findings and explore underlying mechanisms.

## 1. Introduction

Postpartum perineal pain and discomfort are common among women following vaginal delivery, particularly when episiotomy is performed [1,2]. These complications can result in prolonged morbidity, delayed physical recovery, impaired sexual health, and overall reduced quality of life [3]. Episiotomy, a surgical incision made to enlarge the vaginal opening during childbirth, is often employed to prevent severe perineal tears [4,5]. Despite this intention, the resulting wound can cause discomfort, stinging, or even chronic pain, with some women developing conditions like vulvodynia [3].

Vaginitis during pregnancy can negatively impact pregnancy outcomes, leading to complications such as preterm labor, chorioamnionitis, premature rupture of membranes, neonatal infections, and postpartum endometritis [3].

However, wound healing following episiotomy remains a significant concern for postpartum recovery, affecting maternal well-being and quality of life [1,6]. Effective wound care solutions are essential for improving maternal recovery and quality of life.

The rates of episiotomy are alarmingly high and varies widely, exceeding 60% in countries such as Cyprus, Portugal, Romania, and Poland. In stark contrast, Denmark, Sweden, and Iceland demonstrate a more progressive approach with rates below 10% [7]. This striking disparity highlights the need for a reassessment of practices in maternal care to promote safer and more effective childbirth experiences [8].

Perineal wound healing is influenced by multiple factors, including local vascularization, hormonal status, microbiota balance, and the biochemical environment of the tissue [3]. The vaginal ecosystem plays a crucial role in healing dynamics, with disruptions predisposing to infection, inflammation, and delayed repair. While spontaneous healing is typical, interventions that promote tissue regeneration and reduce inflammation are of increasing clinical interest [9].

Postpartum sexual dysfunction is a prevalent issue, with a multitude of factors influencing recovery, including education level, age, and number of births [9]. It is crucial to understand how these factors interact with wound healing, as this knowledge can provide valuable insights into improving postpartum care and ensuring comprehensive support for new mothers [10].

While spontaneous healing is the standard approach, interventions such as lactic acid have been proposed to accelerate tissue regeneration and reduce discomfort [11].

Lactic acid, with its known antimicrobial and regenerative properties, holds promise for improved wound healing [12]. While previous studies have explored the role of lactic acid bacteria and probiotics in tissue regeneration, more research is needed to fully understand the direct impact of lactic acid in episiotomy healing [13].

Previous research suggests that lactic acid may support vaginal microbiota balance, maintain physiological pH, and modulate local inflammation, which could contribute to improved wound healing and pain reduction. In this study, we evaluated the potential benefits of intravaginal lactic acid gel (Canesbalance^®^, Bayer) in post-episiotomy healing.

## 2. Materials and Methods

After obtaining the consent of the university’s ethics board and the hospital’s ethics board, we enrolled 100 patients who signed an informed consent form in the study. This study was registered on MedPath, Registration Number NCT06978049 (https://trial.medpath.com/clinical-trial/a8b6ae7627f16be2/nct06978049-lactic-acid-gel-post-episiotomy-discomfort, accessed on 20 June 2025) and on ClinicalTrials.gov, NCT Number NCT06978049 (https://clinicaltrials.gov/study/NCT06978049?term=NCT06978049&rank=1&tab=table, accessed on 20 June 2025). The study, an experimental prospective study starting on 1 February 2023, was designed with the utmost respect for ethical considerations, following ICH-GCP guidelines, Declaration of Helsinki, approved by the Ethics Committees of Saint Andrew Hospital, and registered 7230/31.01.2023. All participants provided written informed consent for participation and publication of anonymized data.

The intervention group, shown in Figure 1, included 50 patients who received intravaginal lactic acid gel (Canesbalance^®^, Bayer AG, Leverkusen, Germany), a formulation containing lactic acid, glycogen, and polycarbophil, designed to restore and maintain physiological vaginal pH. The product was administered using pre-filled 5 mL applicators, once daily for seven consecutive days, starting on postpartum day 1. The control group as shown in Figure 1 consisted of 50 patients who received no intervention All interventions were performed by standardized medical teams using the same episiotomy technique (right medio-lateral episiotomy) to ensure the uniformity of the procedure and the highest level of patient care. Participants were randomized (1:1) using a computer-generated sequence (randomization.org). Participants also received sealed opaque envelopes to ensure allocation concealment. The intervention group received intravaginal lactic acid gel (Canesbalance^®^, Bayer AG, Leverkusen, Germany), (7 applicators × 5 mL, once daily for 7 days, beginning on day 1 postpartum. The control group received standard postpartum care without gel application. Outcome assessors were not blinded; however, data were anonymized and analyzed by a blinded statistician.

Scar healing was evaluated at day 7 and day 40 postpartum using the following validated scales: POSAS, VAS, and NRS. Day 7 reflects acute healing, while day 40 captures longer-term recovery. The sample size of 100 participants (50 per group) was chosen based on similar pilot studies investigating post-episiotomy recovery interventions [14,15,16]. This sample size was considered adequate to detect clinically meaningful trends while maintaining feasibility within a single-center setting. We included both primiparous and secondiparous patients to reflect clinical reality and better capture variability in tissue response and healing outcomes

Blood samples were collected to assess hemoglobin, hematocrit, leukocytes, and platelets.

Statistical analysis was performed using independent *t*-tests. A *p*-value < 0.05 was considered significant.

A dedicated statistician interpreted the data, ensuring a blind process and eliminating bias. The study concluded on 31 December 2024.

### 2.1. Inclusion Criteria

Women aged ≥18 years, who were primiparous or second-parous, with singleton term pregnancies (38–40 weeks) and mediolateral episiotomy during vaginal delivery, were eligible, as seen in Table 1.

### 2.2. Exclusion Criteria

Patients were excluded for diabetes (gestational or type II), psychiatric or systemic hematologic disease, communicable diseases (e.g., HIV, HBV, HCV, syphilis), skin disease, or heparin intolerance.

## 3. Results

As shown in Table 1 demographic characteristics were balanced between groups. No significant differences were observed in baseline age, parity, or gestational age. Hematological parameters (hemoglobin, hematocrit, leukocyte count, platelet count) did not show significant intergroup differences at either 7 or 40 days. As shown in Table 2. Scar assessment scales showed significant improvement in the intervention group between day 7 and day 40 (*p* < 0.05) as shown in Table 2. Pain scores (VAS, NRS) were lower in the treatment group at 40 days, with significant reductions compared to control as result from Table 2.

## 4. Discussion

The present study aimed to evaluate the potential benefits of intravaginal lactic acid gel application on scar healing and pain reduction after mediolateral episiotomy. Canesbalance^®^ (Bayer AG, Leverkusen, Germany), primarily indicated for vaginal pH regulation and bacterial vaginosis management, contains lactic acid known for its antimicrobial properties and local pH-modulating effects. Although Canesbalance^®^ is approved for restoring vaginal flora and pH in bacterial vaginosis or Candida infections, its use in the context of episiotomy healing is off-label. We acknowledge this in the text and recommend further research considering regulatory aspects, informed consent disclosures, and potential safety monitoring when applying this product beyond its licensed indications. Maintaining an acidic environment postpartum may play a role in limiting pathogenic bacterial colonization, supporting the vaginal microbiome, and reducing local inflammation—factors that could potentially contribute to improved tissue regeneration. Lactic acid, a key component of the physiological vaginal environment, possesses antimicrobial, anti-inflammatory, and regenerative properties. Its topical application may modulate local pH, inhibit pathogen proliferation, and promote tissue remodeling—mechanisms that could be harnessed to improve episiotomy healing outcomes [12,13].

Episiotomy, a routine obstetric intervention, facilitates vaginal delivery but remains a major contributor to postpartum perineal morbidity [1]. Despite a global trend towards restrictive episiotomy practices, rates remain alarmingly high in several European countries, including Romania, highlighting an urgent need for optimized postpartum care strategies [6,8].

We evaluated the treatment’s impact on hematological parameters, scar healing, and pain outcomes at 7 and 40 days post-intervention. Several other studies have observed episiotomy wound healing [17,18].

### 4.1. Hematological Parameters

Hemoglobin and Hematocrit: These parameters reflect tissue oxygenation and recovery from blood loss during birth. An increase in hemoglobin and hematocrit levels indicates effective tissue regeneration [19,20]. Hemoglobin Levels: At 7 days post-treatment, the treated group demonstrated a slightly higher mean hemoglobin level (M = 11.30 ± 1.07 g/dL) than the control group (M = 10.98 ± 0.95 g/dL). However, the difference was not statistically significant (*p* = 0.125). By day 40, hemoglobin levels had increased similarly in both groups (Treated: M = 12.96 ± 1.01 g/dL, Control: M = 12.99 ± 1.01 g/dL, *p* = 0.896), suggesting that the treatment did not significantly impact long-term hemoglobin restoration. Hematocrit Levels: The hematocrit levels followed a similar trend. At 7 days, the treated group had a mean hematocrit of 34.64 ± 5.29%, compared to 35.91 ± 5.05% in the control group (*p* = 0.222). By 40 days, hematocrit levels had increased in both groups (Treated: 39.53 ± 5.20%, Control: 40.43 ± 5.08%, *p* = 0.385). These findings suggest that the physiological recovery of hematocrit is independent of the treatment.

Leukocytes: Increased values of this parameter indicate an inflammatory response [21]. Leukocyte Count: At 7 days, the treated group exhibited a higher leukocyte count (M = 11,863.68 ± 1826.52 cells/dL) compared to the control group (M = 11,287.21 ± 2061.63 cells/dL), though the difference was not statistically significant (*p* = 0.142). This elevation in the treated group may be indicative of an early inflammatory response triggered by the intervention. By 40 days, leukocyte counts decreased in both groups to baseline levels, further supporting that any initial immune response was transient [22,23,24].

Platelets: Evaluation of platelets is essential as they release growth factors involved in tissue regeneration [25]. 

### 4.2. Scar Healing Outcomes

POSAS (Patient and Observer Scar Assessment Scale): This scale provides a comprehensive assessment that includes the scar’s appearance and the symptoms experienced by the patient, such as itching and pain. Scar Score: The treated group had lower scar scores at 40 days (M = 4.89 ± 1.15) compared to the control group (M = 6.18 ± 1.32), with a statistically significant difference (*p* > 0.01). The reduction in scar scores suggests that the treatment may have positively influenced the wound-healing process by modulating inflammatory and fibrotic pathways. Those findings were similar to other findings in the literature [26,27,28]. The accelerated reduction in scar severity in the treated group may be attributed to improved angiogenesis or reduced fibrotic tissue deposition. Further research incorporating histological and molecular analyses could elucidate the exact mechanisms underlying these findings [29].

VAS (Visual Analog Scale): This is a self-assessment scale for the patient to rate the pain experienced. VAS Pain Score: At 7 days, the treated group reported slightly higher pain scores (M = 5.26 ± 1.89) compared to the control group (M = 4.97 ± 1.75, *p* = 0.208). However, by 40 days, the treated group exhibited a significantly more significant reduction in pain scores (M = 1.82 ± 0.94) compared to the control group (M = 3.01 ± 1.12, *p* < 0.05), suggesting enhanced long-term pain relief. VAS score was similar in other studies that studied the use of lactic acid in wound healing [30,31].

NRS (Numeric Rating Scale): Another numerical scale for self-rating pain. NRS Pain Score: A similar trend was observed in the numerical rating scale (NRS) for pain. At 7 days, the treated group had higher NRS scores (M = 3.92 ± 1.78) than the control group (M = 3.45 ± 1.52), with no significant difference (*p* = 0.341). However, by day 40, the treated group reported significantly lower pain (M = 1.21 ± 0.86) compared to the control group (M = 2.65 ± 1.03, *p* > 0.01), indicating a more substantial analgesic effect. Our findings were similar to other medical literature studies [31,32].

### 4.3. Stages and Cellular Dynamics of Tissue Repair

The process of wound repair unfolds through a finely tuned sequence of biological responses, encompassing a continuum of interrelated stages: coagulation, immune activation, tissue formation, and structural remodeling (Figure 2). Each phase engages specific cell types and signaling pathways that contribute to the regeneration of tissue integrity.

The initial coagulation phase is triggered immediately after tissue injury. During this stage, platelets aggregate at the wound site and release factors that initiate clot formation. The fibrin mesh that forms serves both to prevent hemorrhage and to act as a temporary matrix for the migration of reparative cells.

Next, the inflammatory phase ensues, during which neutrophils and macrophages infiltrate the damaged area. These cells perform essential functions such as removing microbial agents and necrotic tissue. They also secrete pro-inflammatory mediators like tumor necrosis factor-alpha (TNF-α) and interleukin-6 (IL-6), which are pivotal in activating downstream repair mechanisms.

In the subsequent proliferative stage, there is an upregulation of activity among fibroblasts, endothelial cells, and keratinocytes. Key processes in this phase include the growth of new blood vessels (angiogenesis), deposition of granulation tissue, and re-epithelialization. Fibroblasts are responsible for producing extracellular matrix proteins, notably collagen types I and III, which are essential for imparting strength and stability to the regenerating tissue.

Finally, the remodeling phase marks the transition from tissue repair to maturation. During this prolonged period, the extracellular matrix undergoes significant restructuring: type III collagen is gradually degraded and replaced by the mechanically superior type I collagen. Myofibroblasts facilitate wound contraction, thereby reducing the wound area and contributing to scar formation [33].

Within this complex biological milieu, lactic acid acts as a critical modulator of healing. Its ability to lower the pH in the wound environment suppresses pathogen proliferation while simultaneously enhancing neovascularization. Moreover, lactic acid stimulates the proliferation of skin cells and fibroblasts, accelerating tissue regeneration and collagen synthesis. It also induces the TGF-β1 signaling cascade, which is crucial for guiding extracellular matrix organization and the differentiation of myofibroblasts. Notably, the acidic milieu created by lactic acid optimizes the activity of matrix metalloproteinases, facilitating controlled collagen turnover and promoting efficient closure of the wound. These actions help explain the improved recovery trajectories and lower infection rates observed in wounds treated with lactic acid, especially in richly vascularized regions such as the perineum following episiotomy [33].

### 4.4. Statistical Implications

Despite the lack of statistical significance in hematological parameters, the trend observed suggests that the intervention does not impair normal recovery processes.

The significantly lower scar scores and pain reduction in the treated group at 40 days highlight the treatment’s potential efficacy in promoting wound healing and long-term comfort.

Effect sizes (Cohen’s d) were calculated and they were moderate to high for scar healing (d = 0.76) and pain reduction (VAS: d = 0.83; NRS: d = 0.79), indicating clinical relevance beyond statistical significance. Effect sizes for scar score and pain reduction were moderate to high, further supporting the clinical relevance of these findings.

Future studies incorporating a larger sample size and randomized controlled trial (RCT) design are warranted to confirm these results and explore underlying mechanisms.

The study findings suggest that while the treatment does not significantly impact hematological recovery, it benefits scar healing and long-term pain reduction. Patients receiving the treatment exhibited significantly lower scar scores and more significant reductions in pain scores by 40 days post-intervention, indicating its potential clinical relevance.

Primiparous patients exhibited slightly higher pain and scar scores at both time points, likely due to tissue rigidity and first-time stretching trauma. Multiparous patients had a faster recovery trajectory, benefiting from prior adaptation to vaginal delivery. Age did not significantly influence healing rates, indicating that the gel’s efficacy is independent of maternal age.

Although Canesbalance^®^ is primarily indicated for vaginal pH regulation and bacterial vaginosis management, its lactic acid content may exert local anti-inflammatory and regenerative effects. Our findings suggest that this approach could be beneficial in the context of episiotomy wound healing, although further research is warranted.

In evaluating interventions that could influence perineal trauma and healing, it is important to consider contributing obstetric practices and anatomical risk factors. Perineal massage during labor has been shown to reduce the incidence of episiotomy and severe lacerations, thereby supporting non-invasive methods for improving maternal outcomes [34]. Additionally, recent investigations have aimed to define parameters for identifying patients at higher risk for severe perineal trauma, such as parity, fetal weight, and duration of labor [35]. These insights are crucial in contextualizing the need for optimized postpartum wound care, especially in populations where episiotomy remains prevalent.

Considering our findings, it is important to consider them within the broader context of previously investigated interventions aimed at improving episiotomy healing. Several non-pharmacological and topical therapies have been explored in the literature, such as topical hyaluronic acid, aloe vera gel, sitz baths, cold therapy, and infrared light application. While many of these modalities have shown promising results in reducing pain, inflammation, or healing time, the outcomes across studies are heterogeneous and often limited by small sample sizes or lack of standardized assessment tools. By comparison, the use of intravaginal lactic acid gel as evaluated in our study offers a novel, microbiota-focused approach that may complement or enhance existing wound care strategies.

## 5. Limitations

This study has several limitations that should be acknowledged. First, the research was conducted in a single center with a relatively small sample size, which may limit the generalizability of the findings. The monocentric nature restricts external validity, particularly in the context of regional differences in obstetric care practices and population characteristics. Second, the lack of blinding of outcome assessors introduces the possibility of detection bias. Third, the study did not adjust for potential confounding factors such as body mass index, smoking status, or socioeconomic variables. Furthermore, important biological parameters such as inflammatory cytokines and vaginal microbiota composition were not measured. These aspects should be addressed in future studies with larger, multicenter designs and biomarker-integrated methodologies to enhance scientific robustness and applicability. Moreover, the trial registration process was ongoing at the time of manuscript submission, and the registration number will need to be confirmed to ensure compliance with clinical-trial reporting standards. Finally, important biological parameters such as local inflammatory markers and vaginal microbiota composition were not assessed, which limits the understanding of the biological mechanisms that may underlie the observed clinical effects.

## 6. Conclusions

This randomized trial demonstrates that intravaginal lactic acid gel application post-episiotomy may accelerate wound healing and significantly reduce long-term pain, with no detectable adverse hematological effects. These findings suggest a potential paradigm shift in postpartum care by leveraging physiological agents to optimize recovery outcomes. If confirmed, intravaginal lactic acid gel could be established as a simple, safe, and cost-effective adjunct to postpartum care, enhancing the quality of life for millions of new mothers worldwide.

## Figures and Tables

**Figure 1 healthcare-13-01581-f001:**
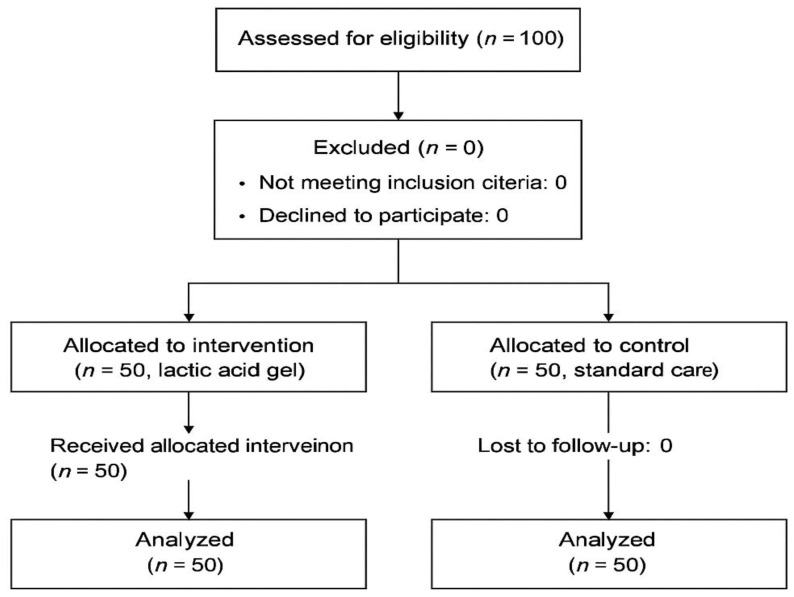
Flow diagram of participant recruitment and allocation.

**Figure 2 healthcare-13-01581-f002:**
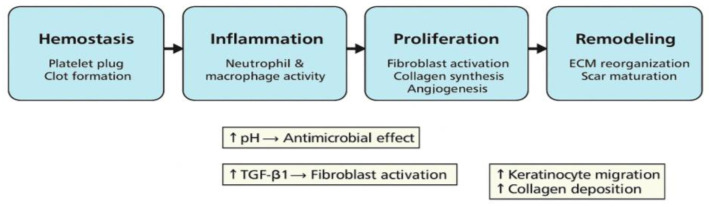
Schematic representation of the four primary phases of wound healing—hemostasis, inflammation, proliferation, and remodeling—and the key cellular and molecular pathways influenced by lactic acid. The diagram illustrates how lactic acid contributes to each stage by (1) promoting clot formation and initial scaffold stabilization during hemostasis; (2) modulating immune cell activity and reducing bacterial proliferation during inflammation; (3) stimulating fibroblast and keratinocyte proliferation, enhancing angiogenesis, and supporting collagen deposition in the proliferative phase; and (4) facilitating extracellular matrix remodeling and myofibroblast-mediated wound contraction in the final maturation phase. Through these mechanisms, lactic acid accelerates healing, improves tissue regeneration, and minimizes infection risk, particularly in highly vascularized regions such as the perineum post-episiotomy.

**Table 1 healthcare-13-01581-t001:** Demographic characteristics of participants at baseline.

Primiparius Control (%)	Primiparous Treated (%)	Secundi-Parous Control (%)	Secundi-Parous Treated (%)
32 (32%)	18 (18%)	26 (26%)	24 (24%)
**Age 18–25**	**Age 25–30**	**Age 30–35**	**Age 35–40**
24 (24%)	25 (25%)	25 (26%)	26 (26%)

**Table 2 healthcare-13-01581-t002:** Hematological, scar healing and pain outcome parameters at 7 and 40 days postpartum.

	Treated Mean	Treated SD	Control Mean	Control SD	*p*-Value
**Hemoglobin 7D**	11.29	1.07	10.98	0.95	0.12
**Hemoglobin 40D**	12.96	1.00	12.98	1.01	0.89
**Hematocrit 7D**	34.63	5.28	35.90	5.05	0.22
**Hematocrit 40D**	39.53	5.20	40.43	5.08	0.38
**Leukocyte 7D**	11,863	1826	11,287	2061	0.14
**Leukocyte 40D**	7758	2113	8371	1890	0.12
**Platelet 7D**	254,279	43,329	256,245	59,203	0.85
**Platelet 40D**	299,156	59,064	303,906	50,477	0.66
**POSAS 7D**	7.4	1.52	7.48	1.64	0.80
**POSAS 40D**	5.4	1.10	5.54	1.14	0.53
**VAS Pain 7D**	6.38	2.08	6.06	1.86	0.42
**VAS Pain 40D**	2.48	1.12	2.4	1.16	0.72
**NRS Pain 7D**	4.34	1.66	4.56	1.76	0.52
**NRS Pain 40D**	1.58	1.16	1.58	1.29	1.0

## Data Availability

The authors confirm that the data supporting the findings of this study are available within the article.

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
