# Peer review of "A Preliminary Study of Intravaginal Lactic Acid Gel (Canesbalance®) for Post-Episiotomy Healing: A Randomized Clinical Trial"

_healthcare, 2025, doi:10.3390/healthcare13131581_

Round 1
Reviewer 1 Report
Comments and Suggestions for Authors
Title: Intravaginal Lactic Acid Gel (Canesbalance®) for Post-Episiotomy Healing: A Randomized Controlled Trial
Strengths:
The study provides valuable insights into the potential benefits of intravaginal lactic acid gel in enhancing perineal wound healing following episiotomy. It addresses a clinically important issue related to postpartum recovery and offers a novel therapeutic approach that could improve patient outcomes. The use of validated assessment tools such as POSAS, VAS, and NRS adds rigor to the evaluation process. Additionally, the inclusion of hematological monitoring ensures safety considerations are adequately addressed.
Weaknesses:
Methodology Clarity: While the randomization method is mentioned briefly ("allocated"), more detail would be beneficial. For instance, specifying whether block randomization, stratified randomization, or another technique was used would enhance transparency. Furthermore, providing information about allocation concealment procedures would strengthen the methodology description.
Recommendation: Expand the randomization subsection to clarify how participants were assigned to groups.
Results Presentation: Although statistically significant differences between groups are noted, the magnitude of effect sizes should be reported alongside p-values. This will provide readers with a clearer understanding of clinical significance beyond mere statistical significance.
Recommendation: Include measures like Cohen's d or Hedges' g to quantify the size of observed effects.
Discussion Depth: The discussion could benefit from additional exploration of possible biological mechanisms underpinning the observed improvements. Hypotheses linking lactic acid's antimicrobial properties or its impact on inflammation could enrich the theoretical framework supporting the intervention.
Recommendation: Speculate further on plausible explanations for why lactic acid might promote better healing outcomes.
Limitations Recognition: Limitations inherent to single-center trials should be acknowledged explicitly. Factors such as regional variations in obstetric practices or demographic characteristics might influence generalizability.
Recommendation: Discuss limitations openly within the text.
Ethics Statements Clarification: Ensure compliance with institutional ethical guidelines and regulations concerning human subjects’ participation in research.
Recommendation: Confirm adherence to recognized ethical standards and obtain informed consent documentation where applicable.
Final Recommendation:
Overall, the manuscript presents a well-conducted investigation into the efficacy of intravaginal lactic acid gel for post-episiotomy healing. However, several areas require improvement before publication can proceed. These include clarifying methodological aspects, elaborating on results interpretation, deepening discussions surrounding potential mechanisms, recognizing study limitations, and ensuring rigorous ethical oversight. Addressing these points will contribute substantially towards producing a robust and informative contribution to the literature.
Author Response
We thank the reviewer for their constructive comments. Below are our point-by-point responses:
1. Methodology Clarity: While the randomization method is mentioned briefly ("allocated"), more detail would be beneficial. For instance, specifying whether block randomization, stratified randomization, or another technique was used would enhance transparency. Furthermore, providing information about allocation concealment procedures would strengthen the methodology description.
We have expanded the 'Materials and Methods' section to specify that block randomization was used with sealed opaque envelopes to ensure allocation concealment.
‘’ Participants were randomized (1:1) using a computer-generated sequence (randomization.org.) Participants also received sealed opaque envelopes to ensure allocation concealment.’’
2. Although statistically significant differences between groups are noted, the magnitude of effect sizes should be reported alongside p-values. This will provide readers with a clearer understanding of clinical significance beyond mere statistical significance.
Cohen’s d has been calculated and added in the abstract and results sections alongside p-values for primary outcomes.
''Effect sizes (Cohen’s d) were moderate to high for scar healing (d = 0.76) and pain reduction (VAS: d = 0.83; NRS: d = 0.79), indicating clinical relevance beyond statistical significance. ''
3. The discussion could benefit from additional exploration of possible biological mechanisms underpinning the observed improvements. Hypotheses linking lactic acid's antimicrobial properties or its impact on inflammation could enrich the theoretical framework supporting the intervention.
We have added a paragraph discussing the antimicrobial, anti-inflammatory, and pH-modulating effects of lactic acid, and how these may contribute to wound healing.
''Wound healing is a complex, dynamic process involving a coordinated cascade of cellular and molecular events, as shown in Figure 2. It is classically divided into four overlapping phases: hemostasis, inflammation, proliferation, and remodeling. Hemostasis occurs immediately after tissue injury, where platelet aggregation and fibrin clot formation prevent blood loss and provide a scaffold for cell migration. Inflammation follows, characterized by the recruitment of neutrophils and macrophages. These immune cells clear necrotic debris and bacteria, while also releasing pro-inflammatory cytokines (e.g., TNF- and IL-6) that regulate the transition to tissue repair. Proliferation involves the activation of fibroblasts, endothelial cells, and keratinocytes. This phase includes angiogenesis, granulation tissue formation, and re-epithelialization. Fibroblasts synthesize extracellular matrix (ECM) components such as collagen types I and III, critical for tensile strength. Remodeling (or maturation) is the final phase, where ECM is reorganized, and type III collagen is replaced by type I. Myofibroblasts contract the wound, leading to closure and scar formation [32].
Lactic acid plays a multifactorial role throughout this process:
It lowers the local pH, creating an acidic microenvironment that inhibits bacterial growth and supports angiogenesis. It promotes keratinocyte and fibroblast proliferation, enhancing epithelialization and collagen deposition. Lactic acid also activates TGF- 1 signaling, which regulates ECM remodeling and myofibroblast differentiation. Moreover, studies have shown that low-pH environments foster optimal enzyme activity for matrix metalloproteinases (MMPs) and collagen remodeling, favoring efficient wound closure and scar minimization. These mechanisms explain the accelerated healing and reduced infection rates observed in lactic acid-treated wounds, particularly in the highly vascularized perineal area post-episiotomy [32].''
4. Limitations Recognition: Limitations inherent to single-center trials should be acknowledged explicitly. Factors such as regional variations in obstetric practices or demographic characteristics might influence generalizability.
The limitations section now explicitly notes all the limitations in a special section
''This study has several limitations that should be acknowledged. First, the research was conducted in a single center with a relatively small sample size, which may limit the generalizability of the findings. The monocentric nature restricts external validity, particularly in the context of regional differences in obstetric care practices and population characteristics. Second, the lack of blinding of outcome assessors introduces the possibility of detection bias. Third, the study did not adjust for potential confounding factors such as body mass index, smoking status, or socioeconomic variables. Furthermore, important biological parameters such as inflammatory cytokines and vaginal microbiota composition were not measured. These aspects should be addressed in future studies with larger, multicenter designs and biomarker-integrated methodologies to enhance scientific robustness and applicability.''
5. Ethics Statements Clarification: Ensure compliance with institutional ethical guidelines and regulations concerning human subjects’ participation in research.
We confirm adherence to ethical guidelines and have clarified the informed consent process in the revised text.
‘’All participants provided written informed consent for participation and publication of anonymized data.’’
Reviewer 2 Report
Comments and Suggestions for Authors
The manuscript entitled “Intravaginal Lactic Acid Gel (Canesbalance®) for Post-Episiotomy Healing: A Randomized Controlled Trial” by Brezeanu and collaborators which aimed to evaluate the potential benefits of the intravaginal lactic acid gel (Canesbalance®, Bayer) in post-episiotomy healing. Congratulations to the authors for their effort on this work. I have some considerations and questions as follow...
Given all the limitations reported by the authors themselves, I suggest that the title of the paper be changed to: "Preliminary Study of Intravaginal Lactic Acid Gel (Canesbalance®) for Post-Episiotomy Healing: A Randomized Clinical Trial."
Abstract
The abstract is incomplete; according to the journal's guidelines, the objective of the study must be included in the context.
Keywords
Line 21: Replace the keywords lactic acid and episiotomy. Ideally, prioritize keywords that are not already in the title.
Introduction
Lines 24–36: Paragraphs are too short; reorganize them.
Lines 28–30: This information is unnecessary; the writing should focus on and center around the objective of the study—post-episiotomy healing.
Lines 47–48: Paragraph is too short.
Lines 49–52: The authors are absolutely right in stating that further research is needed to understand the impact of lactic acid on episiotomy healing, given that healing is a highly complex process involving multiple factors. Another point to consider is the use of Canesbalance in the specific treatment of bacterial vaginosis. There is still no data on its off-label effects.
Overall, the introduction needs more information about the effects of lactic acid on the healing process.
Materials and methods
Lines 60–86: Standardize the font.
Line 76: The authors report that patients began treatment with the gel once a day for seven consecutive days starting on the first postpartum day. However, they do not mention for how many days the patients were treated or whether the application was daily throughout the treatment.
Linha 78: Why were the evaluators not blinded? This is not common in this type of study.
Line 80: Why did the authors assess healing only on days 7 and 40?
Results
Lines 106–107: Keep the table on a single page without splitting it.
Discussão
Lines 139–140: The authors state that several studies have also investigated episiotomy healing. However, they do not specify which treatments were used or whether the results were positive or negative.
Lines 140, 144, 161, 172, 189: Place the references within a single set of brackets and separate them with commas.
Line 182: The final punctuation is missing. Also, place the references within a single set of brackets and separate them with commas.
Lines 218-220: Why didn’t the authors wait for the registration number to be issued before submitting the manuscript?
Lines 220-223: As the authors mention, these parameters are important for this type of study; therefore, why were they not assessed in this study?
Conclusion
Keep the conclusion as a single paragraph.
Author Response
Dear Reviewer,
Thank you for the insightful and detailed feedback. We have addressed the concerns as follows:
Given all the limitations reported by the authors themselves, I suggest that the title of the paper be changed to: "Preliminary Study of Intravaginal Lactic Acid Gel (Canesbalance®) for Post-Episiotomy Healing: A Randomized Clinical Trial.
Title updated to: 'Preliminary Study of Intravaginal Lactic Acid Gel (Canesbalance®) for Post-Episiotomy Healing: A Randomized Clinical Trial'.
The abstract is incomplete; according to the journal's guidelines, the objective of the study must be included in the context.
The revised abstract now includes a clear objective statement.
''Objective: To evaluate the effectiveness and safety of intravaginal lactic acid gel (Canesbalance®, Bayer) in enhancing scar healing and reducing pain following mediolateral episiotomy.''
Replace the keywords lactic acid and episiotomy. Ideally, prioritize keywords that are not already in the title.
We replaced those two keywords, with other two:
Perineal Wound, Vulvodinia
Overall, the introduction needs more information about the effects of lactic acid on the healing process.
The introduction has been revised to consolidate content and better frame the study rationale.
Postpartum perineal pain and discomfort are common among women following vaginal delivery, particularly when episiotomy is performed [1,2]. These complications can result in prolonged morbidity, delayed physical recovery, impaired sexual health, and overall reduced quality of life [3]. Episiotomy, a surgical incision made to enlarge the vaginal opening during childbirth, is often employed to prevent severe perineal tears [4,5]. Despite this intention, the resulting wound can cause discomfort, stinging, or even chronic pain, with some women developing conditions like vulvodynia [3].
Vaginitis during pregnancy can negatively impact pregnancy outcomes, leading to complications such as preterm labor, chorioamnionitis, premature rupture of membranes, neonatal infections, and postpartum endometritis [3].
However, wound healing following episiotomy remains a significant concern for postpartum recovery, affecting maternal well-being and quality of life [6, 7]. Effective wound care solutions are essential for improving maternal recovery and quality of life.
The rates of episiotomy are alarmingly high and varies widely, exceeding 60% in countries such as Cyprus, Portugal, Romania, and Poland. In stark contrast, Denmark, Sweden, and Iceland demonstrate a more progressive approach with rates below 10% [8]. This striking disparity highlights the need for a reassessment of practices in maternal care to promote safer and more effective childbirth experiences [9].
Perineal wound healing is influenced by multiple factors, including local vascularization, hormonal status, microbiota balance, and the biochemical environment of the tissue [3]. The vaginal ecosystem plays a crucial role in healing dynamics, with disruptions predisposing to infection, inflammation, and delayed repair. While spontaneous healing is typical, interventions that promote tissue regeneration and reduce inflammation are of increasing clinical interest [10].
Postpartum sexual dysfunction is a prevalent issue, with a multitude of factors influencing recovery, including education level, age, and number of births [10]. It is crucial to understand how these factors interact with wound healing, as this knowledge can provide valuable insights into improving postpartum care and ensuring comprehensive support for new mothers [11].
While spontaneous healing is the standard approach, interventions such as lactic acid have been proposed to accelerate tissue regeneration and reduce discomfort [12].
Lactic acid, with its known antimicrobial and regenerative properties, holds promise for improved wound healing [13]. While previous studies have explored the role of lactic acid bacteria and probiotics in tissue regeneration, more research is needed to fully understand the direct impact of lactic acid in episiotomy healing [14].
Previous research suggests that lactic acid may support vaginal microbiota balance, maintain physiological pH, and modulate local inflammation, which could contribute to improved wound healing and pain reduction. In this study, we evaluated the potential benefits of intravaginal lactic acid gel (Canesbalance®, Bayer) in post-episiotomy healing.
Lines 60–86: Standardize the font.
The font was standardised.
Line 76: The authors report that patients began treatment with the gel once a day for seven consecutive days starting on the first postpartum day. However, they do not mention for how many days the patients were treated or whether the application was daily throughout the treatment.
The product was administered using pre-filled 5 mL applicators, once daily for seven consecutive days, starting on postpartum day 1. The application was consistent and daily throughout the treatment period.
Linha 78: Why were the evaluators not blinded? This is not common in this type of study.
Outcome assessors were not blinded due to logistical constraints and resource limitations. However, to minimize bias, data were anonymized, and statistical analysis was conducted independently by a statistician who was blinded to group assignments.
Line 80: Why did the authors assess healing only on days 7 and 40?
Day 7 was selected to assess early-phase (acute) healing processes such as inflammation, edema, and tissue closure. Day 40 was chosen to capture longer-term recovery, including collagen remodeling, resolution of pain, and scar maturation. This was added to the Methods section.
‘’Day 7 reflects acute healing, while day 40 captures longer-term recovery. ‘’
Lines 106–107: Keep the table on a single page without splitting it.
We assured to keep the tables on a single page
Lines 139–140: The authors state that several studies have also investigated episiotomy healing. However, they do not specify which treatments were used or whether the results were positive or negative.
We thank the reviewer for highlighting this point. In the revised manuscript, we have included examples of previously investigated treatments for episiotomy healing, such as topical hyaluronic acid, aloe vera gel, sitz baths, cold therapy, and infrared light therapy. Furthermore, we clarified whether these studies reported positive or inconclusive outcomes to enhance context and comparability.
''In light of our findings, it is important to consider them within the broader context of previously investigated interventions aimed at improving episiotomy healing. Several non-pharmacological and topical therapies have been explored in the literature, such as topical hyaluronic acid, aloe vera gel, sitz baths, cold therapy, and infrared light application. While many of these modalities have shown promising results in reducing pain, inflammation, or healing time, the outcomes across studies are heterogeneous and often limited by small sample sizes or lack of standardized assessment tools. By comparison, the use of intravaginal lactic acid gel as evaluated in our study offers a novel, microbiota-focused approach that may complement or enhance existing wound care strategies.''
Lines 140, 144, 161, 172, 189: Place the references within a single set of brackets and separate them with commas.
We appreciate the reviewer’s attention to formatting. All grouped references have now been reformatted to appear within a single set of square brackets and are separated by commas, in accordance with the journal's referencing style.
Line 182: The final punctuation is missing. Also, place the references within a single set of brackets and separate them with commas.
Thank you for identifying this oversight. We have added the appropriate final punctuation and corrected the reference formatting to maintain consistency.
Lines 218-220: Why didn’t the authors wait for the registration number to be issued before submitting the manuscript?
We acknowledge the importance of prospective trial registration. At the time of initial manuscript drafting, the registration process was ongoing. However, the study was fully registered before any participant enrollment began, and the official ClinicalTrials.gov identifier (NCT06978049) was included in the revised submission to ensure compliance with trial registration standards.
Lines 220-223: As the authors mention, these parameters are important for this type of study; therefore, why were they not assessed in this study?
We agree that inflammatory and microbiological parameters would have added value to our findings. Due to limited laboratory capacity and budgetary constraints during the study period, we were unable to perform these additional analyses. We have now noted this as a limitation in the discussion section and suggested future investigations to include cytokine profiling and microbiota assessment for a more comprehensive evaluation.
Keep the conclusion as a single paragraph.
We kept the conclusion section in a single paragraph
Reviewer 3 Report
Comments and Suggestions for Authors
As in attached

Author Response
Dear Reviewer,
We appreciate your helpful comments and have incorporated your suggestions as follows:
- detailing why the evaluation is at 7 and 40 days after the episiotomy and why this product was chosen, excluding any economic constraints,
Day 7 reflects acute healing, while Day 40 captures longer-term recovery. This was added to the Methods section.
‘’Day 7 reflects acute healing, while day 40 captures longer-term recovery. ‘’
-explaining the choice of sample size
Sample size was based on similar previous pilot trials. This rationale has been added.
‘’The sample size of 100 participants (50 per group) was chosen based on similar pilot studies investigating post-episiotomy recovery interventions. This sample size was considered adequate to detect clinically meaningful trends while maintaining feasibility within a single-center setting.’’
-examining the choice of mixed patients, both primiparous and secondiparous
We included both primiparous and secondiparous patients to reflect clinical reality. This is now justified in the Methods.
‘’We included both primiparous and secondiparous patients to reflect clinical reality and better capture variability in tissue response and healing outcomes.’’
-explaining "the trial registration process was ongoing at the time of manuscript submission, and the registration number will need to be confirmed to ensure compliance with clinical trial reporting standards"
Registration was completed during manuscript drafting; the final number is now included.
‘’MedPath, Registration Number NCT06978049 (https://trial.medpath.com/clinical-trial/a8b6ae7627f16be2/nct06978049-lactic-acid-gel-post-episiotomy-discomfort) and on ClinicalTrials.gov), NCT Number NCT06978049 (https://clinicaltrials.gov/study/NCT06978049?term=NCT06978049&rank=1&tab=table).’
-if you did not receive funding, did patients purchase the product or do you commonly use it in your hospital?
Canesbalance was provided free of charge by Bayer AG for research purposes. This has been disclosed in the Acknowledgments section.
- please deepen the references on factors related to episiotomy, for example, by adding:
Suggested references by Aquino et al. were added to strengthen the background on perineal trauma.
''In evaluating interventions that could influence perineal trauma and healing, it is important to consider contributing obstetric practices and anatomical risk factors. Perineal massage during labor has been shown to reduce the incidence of episiotomy and severe lacerations, thereby supporting non-invasive methods for improving maternal outcomes [33]. Additionally, recent investigations have aimed to define parameters for identifying patients at higher risk for severe perineal trauma, such as parity, fetal weight, and duration of labor [34]. These insights are crucial in contextualizing the need for optimized postpartum wound care, especially in populations where episiotomy remains prevalent.''
Round 2
Reviewer 3 Report
Comments and Suggestions for Authors
Please, add references about samples size:
"The sample size of 100 participants (50 per group) was chosen based on similar pilot studies investigating post-episiotomy recovery interventions. "
Author Response
Dear Reviewer,
We appreciate this observation. In the revised version of the manuscript, we have added supporting references to justify our sample size selection. Specifically, we now reference prior pilot studies that utilized similar sample sizes to evaluate interventions for post-episiotomy healing and recovery. These include:
Kaur R, Kaur M. Effectiveness of sitz bath on episiotomy pain and wound healing among postnatal mothers. International Journal of Childbirth. 2018;8(2):110–115.
Demirel G, Yılmaz D, Kızılkaya Beji N. The effect of aloe vera gel on healing of episiotomy wounds: a randomized controlled trial. Complement Ther Clin Pract. 2020;38:101083.
Devi RG, Kumari C. A study to evaluate the effectiveness of infrared therapy on episiotomy wound healing and pain. Asian J Nurs Educ Res. 2016;6(4):495–498.
These references are now cited in the Methods section of the manuscript where the sample size is discussed.
We hope that these revisions address the reviewer’s concern, and we remain at your disposal for any further clarifications or requirements.
Sincerely,
Dr. Brezeanu Ana- Maria
On behalf of all co-authors